behaviour/psychology/cognition

social science, laboratory in the field, cross-cultural comparisons, methods, experiments, replication

**Authors for correspondence:**
Anne C. Pisor
e-mail: anne.pisor@wsu.edu
Cody T. Ross
e-mail: cody_ross@eva.mpg.de

# Preferences and constraints: the value of economic games for studying human behaviour

Anne C. Pisor[1,2], Matthew M. Gervais[3],
Benjamin G. Purzycki[2,4] and Cody T. Ross[2]

[1]Department of Anthropology, Washington State University, Pullman, WA 99164-4910, USA
[2]Department of Human Behavior, Ecology, and Culture, Max Planck Institute for Evolutionary Anthropology, Deutscher Platz 6, 04103 Leipzig Germany
[3]Centre Culture and Evolution, Brunel University, London, UK
[4]Department of the Study of Religion, Aarhus University, Aarhus, Denmark

ACP, 0000-0001-5780-4542; MMG, 0000-0002-2532-2722;
BGP, 0000-0002-9595-7360; CTR, 0000-0002-0067-4799

As economic games have spread from experimental economics to other social sciences, so too have critiques of their usefulness for drawing inferences about the 'real world'. What these criticisms often miss is that games can be used to reveal individuals' private preferences in ways that observational and interview data cannot; furthermore, economic games can be designed such that they *do* provide insights into real-world behaviour. Here, we draw on our collective experience using economic games in field contexts to illustrate how researchers can strategically alter the framing or design of economic games to draw inferences about private-world or real-world preferences. A detailed case study from coastal Colombia provides an example of the subtleties of game design and how games can be combined fruitfully with self-report data. We close with a list of concrete recommendations for how to modify economic games to better match particular research questions and research contexts.

## 1. Introduction

Over the last three decades, economic games have become an important part of the toolkits of social scientists. Originating in experimental economics in the 1970s and 1980s [1], researchers initially designed economic games to isolate particular features of real-world rules or norms, such as those observed in bargaining, in order to compare individuals' preferences in the

game to those predicted by economic theory [2]. Economists have a high bar for establishing causality, so in order to investigate the causal relationship between experimental[1] manipulations and participant behaviour, economists designed their games to be highly internally valid—that is, standardized and repeatable [7–9]. Given this initial emphasis on internal validity, it is perhaps unsurprising that economic games have been criticized in economics for their lack of external validity, or generalizability to situations beyond that of the experiment [7,10]. As economic games have made their way into the toolkits of other disciplines, such as anthropology and psychology, this criticism has become more widespread [11–13]. Critics have argued that the rules and norms that inspired classical economic games may not apply outside of large-scale market economies [12,13] or be relevant to theoretical questions beyond bargaining [7,11]. More generally, there is an increasing consensus that researchers should not use the standard suite of economic games *just* because they are 'standardized' [9] or 'paradigmatic' [9,14].

Abandoning economic games as a methodological tool on the basis of these critiques, however, would be a mistake. Economic games are useful precisely because researchers can tailor the experimental design of these games to reflect relevant features of the real world, including those specific to a local context [15,16]. Furthermore, researchers can design games such that they minimize (though not eliminate) real-world constraints—for example, those posed by resource limitations (individuals simply may not have extra resources to give to others) or the reputational consequences of real-world behaviour (individuals may have preferences to be stingy that they are unable to express in daily life)—revealing participants' private preferences in a way that observational and interview data may not. In the present paper, we argue for the extended use of economic games alongside other social science methods. We focus on the potential of modified game designs, and we review studies that we have conducted in collaboration with societies around the globe. We describe in detail a case study from Colombia that demonstrates both how experimental data can be an important *complement* to observational and self-report data [17] (cf. [10]) and why careful experimental design is key to making inferences about either private-world or real-world preferences. We close with suggestions for how researchers can modify classical economic games to better reflect their research questions and the features of the real world relevant to their studies.

## 1.1. Validity in what external context?

In most economic games, researchers provide participants with money—ranging from the equivalent of a few minutes' work to a day's wages or more [18]—and ask them to decide how much to keep for themselves and how much to allocate to third parties (the *recipients*). Under some experimental designs, such as that of the Ultimatum Game (see [14] for a review), recipients can respond to the decisions made by the focal participant (the *decider*); under others, such as that of the Dictator Game (see [9]), recipients cannot respond (see figure 1 for an illustration). Decisions are usually made in private, without the researcher watching, and recorded afterwards. Deciders and recipients usually receive their payouts immediately.

Many critiques of experimental games claim that they fail to reflect important features of the real world—that is, they lack ecological validity (see [19] for a discussion). These critiques, especially those originating in anthropology and psychology, usually take one of two forms. The first focuses on the concept of anonymity in classical economic games. In an effort to minimize concerns with reciprocation or reputation, classical games like the Ultimatum and Dictator Games are typically played anonymously, so that deciders and recipients do not learn each other's identity [1,20]. Some critics argue that in small towns or in subsistence-scale societies where interactions with anonymous others are rare, this aspect of game design does not reflect participants' daily lives [21]. Furthermore, participants know that only a subset of their community is playing the game, and that their fellow players must be among these individuals, undermining researchers' attempts to emulate anonymity.

The second critique of the ecological validity of economic games concerns how games are described to participants. To avoid biasing deciders' decisions, many classical economic games present participants

---

[1]Though not all economic game studies are 'experiments' in the strictest sense, involving the manipulation of an independent variable [3], in principle games are experimental because they afford such manipulation. For example, researchers can test the effects of punishment on generosity [4] or compare anonymous and non-anonymous treatments (see relevant discussion in §2.2). Even without different treatments, economic games fit the less-stringent definitions of 'experiments' in areas of psychology (see discussion in [5]) and in experimental economics [6]. As such, we follow custom in experimental economics and refer to games as experiments here.

**Figure 1.** The basic structure of the Dictator and Ultimatum Games. For further details about each of these classical games, see [9,14]. Images by Gordon Johnson and OpenClipart-Vectors from Pixabay.

with minimalistic descriptions of the task, avoiding real-world analogies or terms like 'donation' or 'gift' [22]. As in most experiments, the results generated by economic games are sensitive to the framing of the experiment—the way researchers describe and present the experiment to participants [23,24]. Results from games framed with real-world analogies can differ substantially from those without such framing [24–26]. When instructions are minimalistic, it is unclear what real-world analogies or lived experiences participants draw on to guide their decision-making [11,12,16,27,28], muddying interpretations of the data.

Concerns about how anonymity and framing impact economic game play are not unfounded, but these are limitations intrinsic to many experiments, not only to economic games. The ecological validity of games hinges entirely on the contexts to which the results are meant to map. In other words, what is meant by the 'real world' [6]? Does the 'real world' refer to behaviour in market interactions? To behaviour in houses of worship? To behaviour in situations requiring cooperative labour? Just as we should not expect observational data collected in one context to generalize to another context—a central tenet of cultural anthropology—we should not expect data from one experimental context to generalize to an untold number of real-world contexts [22,29]—a central tenet of cross-cultural psychology. Experimental studies are normally designed to evaluate theory [15], not mimic the real world.

Critiques of the ecological validity of economic games do suggest that games must be used with explicit consideration of their affordances—not just because they are 'standardized' or 'paradigmatic'. This includes consideration of: (i) whether the game design can address a particular research question, (ii) whether the research goal is inference about preferences under minimal constraints or about preferences under real-world constraints, and (iii) whether the game will be understandable to, and conceptualized similarly by, all participants. For example, behaviour in anonymous games may reveal an individual's cooperative preferences [30] absent concern for recipient characteristics (e.g. how nice they seem) or reciprocal obligations (e.g. 'I owe them one') and may plausibly reflect local norms for behaviour during interactions with strangers [31]. By contrast, games in which the decider is anonymous but the *recipient* is not may reveal decider preferences conditional on recipient characteristics and relational characteristics (e.g. past disagreements between the decider and the recipient), revealing how cooperative behaviour is structured by interpersonal sentiments [16]. Furthermore, games in which neither the decider *nor* the recipient are anonymous may reveal decider preferences to make a positive first impression on a stranger [32,33].

In the next section, we demonstrate how anonymity and experimental framing can strategically be modified to reveal different participant preferences, with illustrations from our own experimental

work. We focus separately on the measurement of private-world preferences—what participants would do given minimal constraints—and on measuring preferences in real-world relationships.

# 2. Altering classical games to answer new questions

## 2.1. A private world: measuring preferences in the context of minimal constraints

The notion of 'preferences', or the (often ranked) importance people give to different things or actions, can provide insights into the nature of human decision-making—even a window into the evolutionary history of human decision-making [23,34,35]. This is not to say that these preferences are independent of cultural influence [36]: many factors may contribute to an individual's preferences, including their sentiments—their attitudes and emotions towards particular people [37] (see also [38,39]); their assets, such as their perceived socio-economic status [32] and their housing and food security [40]; and values they have acquired from social transmission and internalized, such as moral culture (notions of what makes other people good or bad [41]). Importantly, individuals cannot always act according to their preferences in real life due to constraints on their behaviour from cultural institutions and social obligations (e.g. when money is requested, one must share [42,43]). In other words, social structure constrains an individual's agency with consequences, foregrounding some preferences and masking others [44]. Because of this, researchers may have difficulty using observational or self-report data to study participant preferences with minimal constraints—i.e. their private-world preferences. Observational data generally reflect only what individuals are able to do given real-life constraints, such as resource and time availability, the expectations of family or community members, and institutional sanctions. Furthermore, social desirability concerns can colour self-report data. Participants may not wish to reveal socially unacceptable preferences to the researcher—for example, it is much cheaper to say that one always shares than it is to forgo a half day's wages to do so.

Experimental techniques, including economic games, are a useful tool for reducing the effects of external constraints on behaviour and attenuating social desirability biases, improving our insight into private preferences—how participants would behave if they could. Allocation games like the Dictator Game are particularly good at capturing private preferences, because they give deciders more agency than do games that allow recipients to respond [9,45,46]: when making allocation decisions, deciders do not need to use their own resources, acquired outside the experimental context [47], nor anticipate how recipients [48], third parties or the researcher will react to their decisions [20,49].

B.G.P. and A.C.P. have used two different allocation games to measure deciders' preferences with respect to recipients with whom they only rarely interact. While rare interactions are difficult to capture with observational data—as they may not take place during a field season—or with interview data—as they may not be salient enough in the context of the interview to be recalled and reported by participants [50,51]—experimental methods can allow researchers a window into these infrequent encounters. Furthermore, both B.G.P. and A.C.P. asked participants to make their allocation decisions in private to minimize the risk of self-presentation bias.

B.G.P. and colleagues used what they call a random allocation game (RAG) [40] to test whether moral values and belief in morally concerned deities affect rule-following [41,52–56]. Deciders played two games. In one game, participants were presented with two cups: one representing an anonymous same-community co-ethnic, co-religionist individual, and one representing a geographically distant co-ethnic, co-religionist individual. In the other game, the cups represented the decider (self) and a separate co-ethnic, co-religious individual from the same distant community. Researchers presented deciders with 30 coins and a fair, six-sided die with three sides of one colour and three sides of another. They asked deciders to mentally choose a cup and a colour, then to roll the die 30 times in private, without the researcher watching; if the chosen colour was rolled, deciders were told to put a coin into their chosen cup, and if the other colour was rolled, they were told to put the coin into the other cup (figure 2). If deciders did not break game rules to put more coins in one cup instead of the other, half of the 30 coins would end up in each cup on average. Because decisions were made in private and recipients could not respond, deciders could act according to their private preferences to either favour their local community or to allocate fairly. The RAG allowed researchers to examine the role of religion—and, more specifically, belief in morally concerned, punitive deities—in widening the sphere of human cooperation to include geographically distant, co-religionists as if they were members of one's local community.

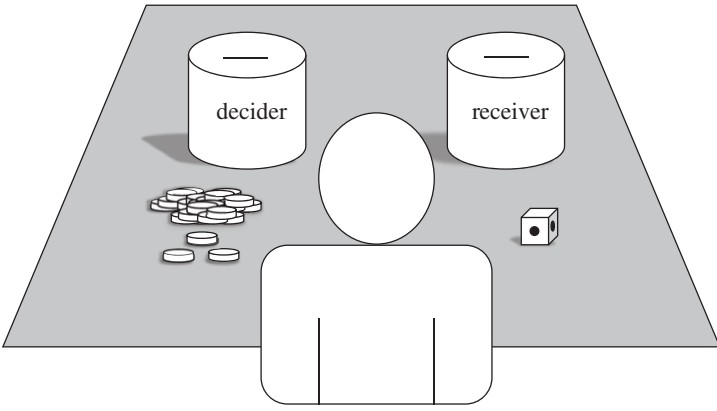

**Figure 2.** Set-up for the RAG including coins, cups and a fair, two-coloured die (illustrated here with spots instead of colours). All money left in each cup was distributed to one randomly selected individual described on that cup (e.g. someone from the same community, same religion and same ethnic group).

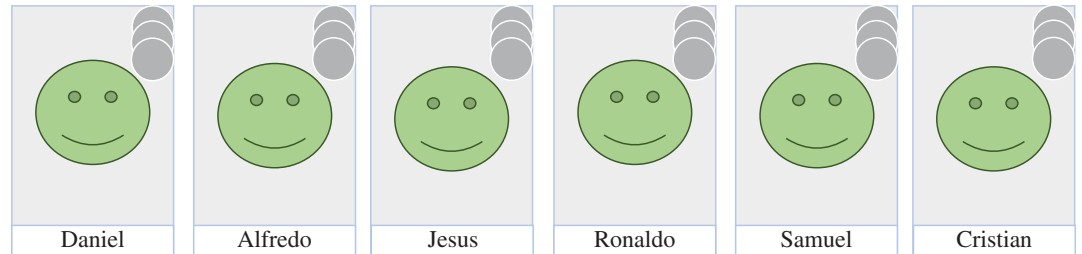

**Figure 3.** Set-up of the experiment in Bolivia, with the simultaneous presentation of photos of ingroup and outgroup strangers (e.g. ingroup, three on left; outgroup, three on right) and initial allocation of coins (decider was also allocated three coins, not pictured) [33]; reprinted under CC-BY-NC-ND.

Studying individuals' preferences for outgroup versus ingroup relationships, A.C.P. and Michael Gurven used an experimental paradigm in which recipients were strangers, but members of either the decider's ethnolinguistic/religious group or a different ethnolinguistic/religious group [33]. Members of one of the three Bolivian populations with whom A.C.P. collaborates, the Tsimane', have minimal access to roads or to cheap river transportation; because of this, Tsimane' individuals have only infrequent interactions with members of other ethnolinguistic groups [32], making these interactions difficult to capture via observational study designs. Furthermore, when intergroup interactions *do* occur, the Tsimane' often self-report suffering discrimination [57]. Given the rarity of intergroup interactions, A.C.P. attempted to approximate a first-time interaction across group boundaries using a non-anonymous allocation game: deciders were simultaneously presented with photos of ingroup and outgroup strangers and learned the name and group membership of each (figure 3); recipients learned the first and last name and group membership of those who gave them money. Tsimane' deciders were 60% less likely to give a coin to a recipient from a group they perceived as having good market access, allocating more money to other Tsimane' recipients instead [33]. In post-game interviews, deciders frequently indicated that they preferred to allocate coins to recipients who were in greater need, consistent with the common view among the Tsimane' that the Tsimane' have fewer resources than other ethnolinguistic groups in Bolivia [33]. A study based only on observational and self-report data might simply attribute the infrequent interactions between the Tsimane' and other ethnic groups to discrimination or lack of mobility. While both of these factors may play a role, this experimental paradigm reveals that Tsimane' preferences may also be influenced by need, such that they prefer to channel money towards those they feel need it most—other Tsimane'.

## 2.2. A real world: measuring preferences in the context of real-world constraints

In a given study, social scientists may not want to know about individuals' preferences with minimal influence from real-world constraints—how they would behave if they could—but instead may want to know about individuals' preferences *given* real-world constraints. In the latter paradigm,

participants' behaviour in experiments is treated as a proxy for how they behave in the real world. For example, economic games are useful for testing the assumptions of applied interventions (e.g. establishing a protected fishery) before or during project implementation [58,59]. Games permit inferences about how individuals behave in light of local cultural institutions, like those governing interactions with strangers [31,60] or with members of other ethnic groups [48,61–63]. Games can also reveal how an individual's preferences reflect the qualities of their social partners. Studies like these— what we call the real-world approach—are unfortunately often conflated with the private-world approach: just as private-world research is critiqued for not approximating the real world (§2.1), real-world research is critiqued for inadequately testing theory (see [8] for discussion), even though that is not what it is designed to test.

To understand how individual characteristics structure real-world social relationships on the island of Yasawa in Fiji, M.M.G. developed three recipient identity-conditioned heuristics (RICH) games [16]. In these games, researchers present deciders with a photo array of same-community members (figure 4). In an allocation game, researchers give deciders coins and deciders must choose how to allocate these coins across recipients, including themselves; in a taking game, researchers distribute coins across recipient photos and deciders must choose whether or not to take coins from the recipients; and in a costly reduction game, deciders are given coins and must choose whether or not to pay to reduce the total amount a recipient receives. RICH games reveal how a decider's preferences to give to, exploit or punish a particular recipient reflect the decider's characteristics (e.g. their resources), the recipient's characteristics (e.g. their reputation) and properties of the dyad (e.g. kinship and friendship).

When playing these games in Fiji, almost all participants provided rationales for their decisions that were consistent with M.M.G.'s observational work, including a desire to help the 'weak' and punish 'moneyheads'. Furthermore, parallels between the games and the real world were not lost on participants. In debriefing interviews, participants were two-to-three times as likely to relate RICH games to their daily lives as were participants in neighbouring communities who played classical anonymous games [64]. Multi-level analyses of these data [65] reveal large effects of recipient reputations, dyadic relationships and interpersonal sentiments on game decisions, factors that drive everyday village decision-making (e.g. [66]) but are masked in anonymous-recipient games. Consequently, participants displayed levels of both generosity and punitiveness not observed in anonymous-recipient games; as decisions were confidential, punitiveness also exceeded that expected from ethnographic observations.

While designed to tap nominally distinct motives to help (the allocation game), exploit (the taking game) and punish at a cost (the costly reduction game), the different RICH games also differ subtly in their parameters. For example, in the allocation game, deciders do not have enough coins to give to all recipients, but in the taking game, deciders can choose to leave already-distributed coins on every photo, so that *all* recipients receive some. In other words, the games entail different degrees of resource constraint. Such subtle differences in experimental design can affect the degree to which participants can exhibit real-world or private-world preferences. We illustrate this below with RICH games data C.T.R. collected in Colombia.

# 3. A private and a real world: ethnicity, wealth, and food insecurity in Colombia

## 3.1. Research context

C.T.R. conducts research with an artisanal fishing community of Afrocolombians and Emberá on the Pacific coast of Colombia. Nearly all of the Emberá and a large proportion of the Afrocolombians in the community are considered internally displaced persons, affected by Colombia's internal conflicts. In the region, a majority of residents are Afrocolombian (82%), followed by Emberá and related groups (13%), and a small fraction of Mestizos (5%) [67]. Inequality is high in the community, both in terms of reported income (Gini = 0.47) and material wealth (Gini = 0.40), with poorer individuals residing on lower-quality land (e.g. on the borders of landfills or on tidal lands). Subsistence for the Afrocolombian community is based around artisanal fishing, whereas horticulture forms the basis of Emberá subsistence. Regardless of the ethnic group membership, however, wage labour, hunting, fishing, horticulture and animal husbandry may all be practised.

Social network questionnaires reveal that most individuals interact primarily within cliques [68] composed of members of their own ethnic group and their neighbours (figure 5). Social relationships

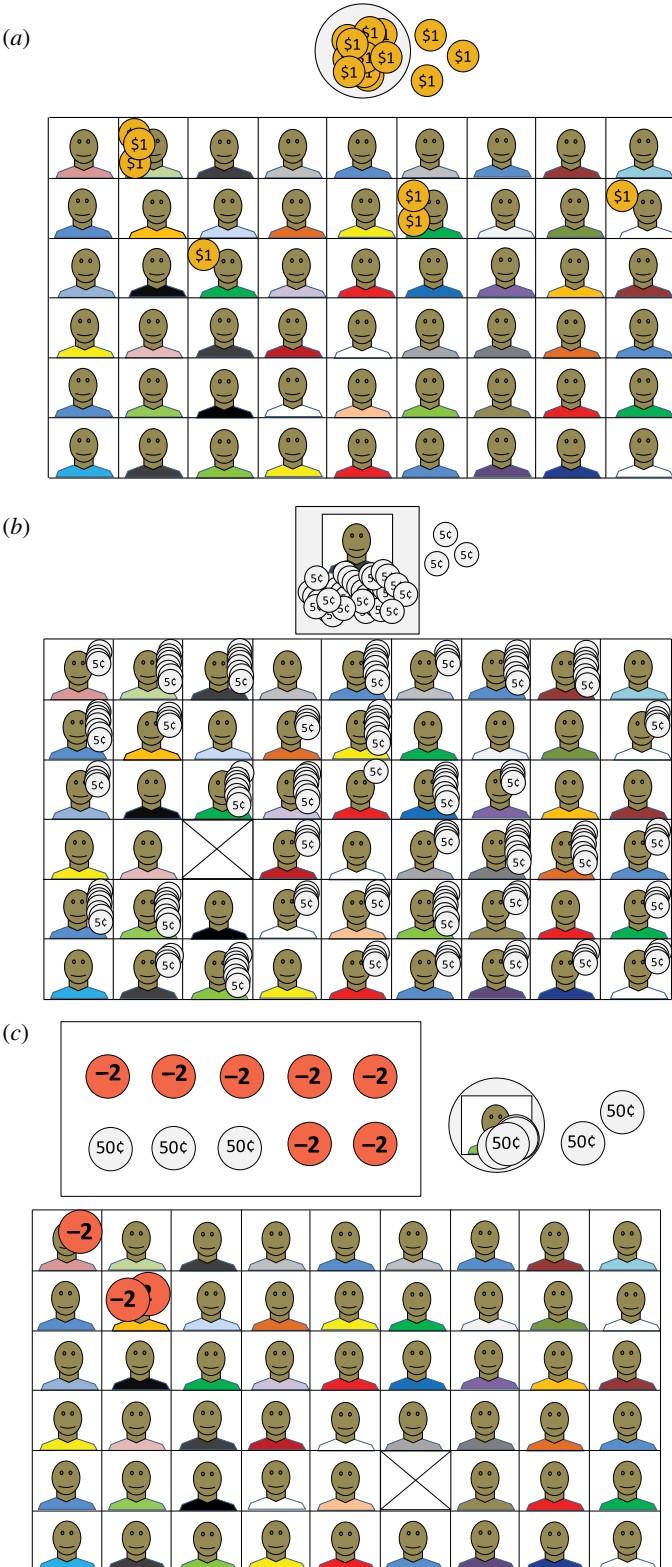

**Figure 4.** Illustrations of decisions in progress for the (*a*) allocation, (*b*) taking, and (*c*) costly reduction RICH games; photos are of same-community individuals. Adapted from [16].

help individuals buffer the resource shocks associated with poverty and the resettlement resulting from forced displacement. Reported resource sharing networks are similar in structure to friendship networks, indicating that giving is generally structured by social closeness, kinship and distance between households (figure 5*a*,*b*). Note, however, that a fraction of interethnic resource transfer ties may have

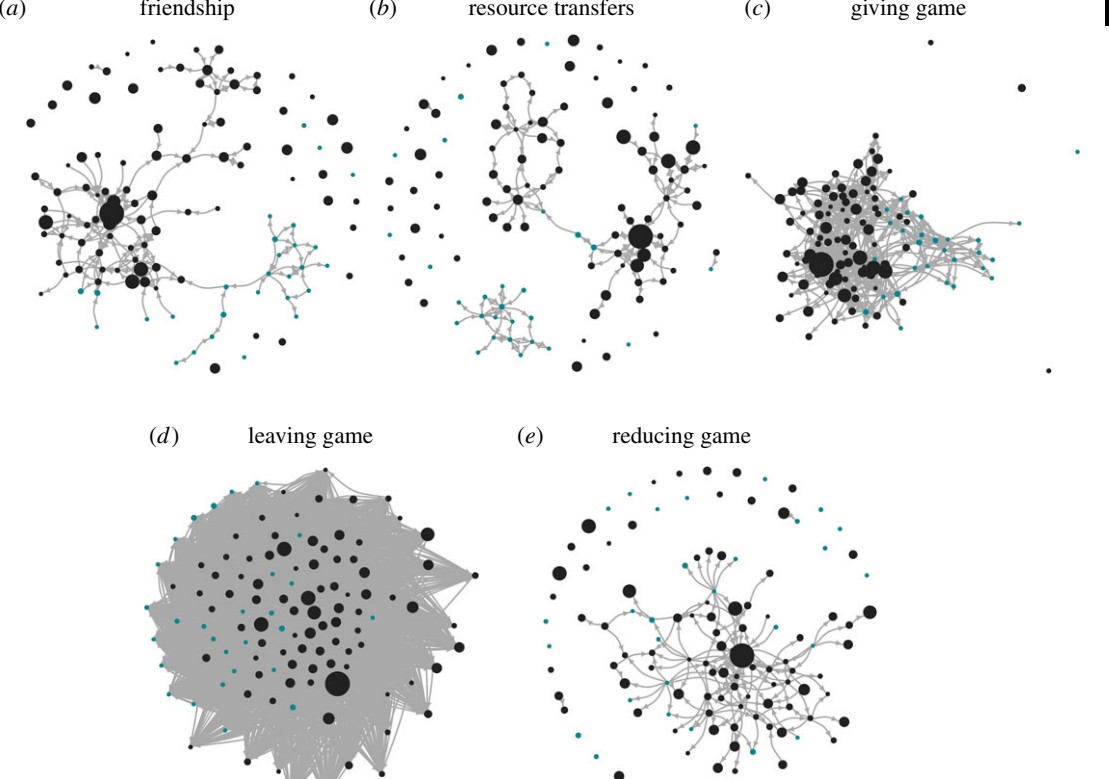

**Figure 5.** Network structure of (*a*) social relationships, (*b*) resource transfers, and (*c*–*e*) RICH game data. Afrocolombians are plotted in black and Emberá in blue. Points (also called vertices) represent unique individuals and are scaled by material wealth. Ties (also called edges) represent social connections or resource/experimental transfers between a pair of individuals; both unidirectional and bidirectional ties are depicted. Frame (*a*) illustrates the social relationships between individuals and illustrates the role of ethnicity and wealth similarity in the structure of these relationships. Frame (*b*) shows the flow of food and money transfers and illustrates a rather low density of connections. In frame (*c*), a higher density of transfers in the RICH game becomes apparent, as every decider has money available to give should they choose to. Giving remains more common between dyads of the same ethnicity. In frame (*d*), deciders have enough coins to allocate to all recipients, and the resulting density of transfer ties increases. The abundance of ties reflects that most deciders left money for (i.e. failed to take money from) most recipients. Finally, in frame (*e*), we see that comparatively few deciders paid to reduce recipients.

been missed; some Afrocolombians reported making small transfers of food or money to Emberá community residents who asked for these transfers, but whose names the Afrocolombians did not know (cf. Figure 5*b*).

## 3.2. RICH game methods

In 2017, C.T.R. administered the RICH game protocols described in §2.2., including the allocation game, taking game and costly reduction game [16], to 93 individuals (54% female; 76% Afrocolombian). Recipients were individuals from the same community, including members of the decider's own household. He then paired RICH game data with demographic, anthropometric (i.e. body measurement) and social network data collected in 2016, and designed statistical models to predict a decider's behaviour with respect to each candidate recipient. Independent models were used to predict decider behaviour in each of the three economic games and in a real-world food/money transfer network (described in §3.1). To model the zero-sum nature of economic games with multiple recipients (see also [33]), C.T.R. used Bayesian mixed-effect multinomial regressions similar in structure to the social relations model used by Koster & Leckie [47]. Analyses were coded in Stan [69] and implemented with the R statistical program (v. 3.6.0, [70]) using the RStan package (v. 2.18.2, [71]). Results are reported as the mean of the posterior distribution with 90% credible intervals, akin to confidence intervals. In the interest of space, further methodological details and robusticity checks

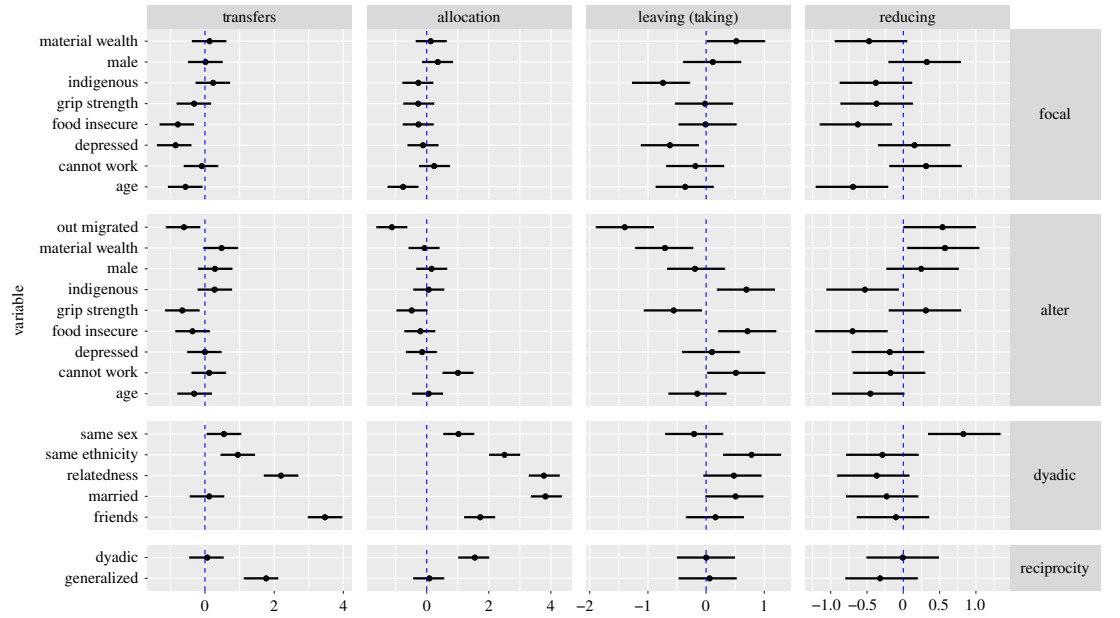

**Figure 6.** Standardized coefficient estimates for the effects of focal, recipient and dyadic characteristics on the probability of a focal individual making an outgoing transfer. Points show posterior medians, and error bars show 90% posterior credible intervals.

can be found in the electronic supplementary materials. The code and data used are available at www.github.com/ctross/preferencesandconstraints.

## 3.3. Real-world and private-world preferences

A comparison of model estimates for the predictors of resource transfers (figure 6, column 1) with those of game play in the three RICH games (figure 6, columns 2–4) demonstrates both the parallels and key differences between real-world behaviour and game play. With respect to the parallels, note first the dyadic measures of perennial interest to social scientists: ethnic affiliation, kinship, friendship, and reciprocation of cooperative behaviour (rows 3–4). For these predictor variables, we observe substantial agreement between the effects detected in the allocation game and the effects detected in the real-world resource transfer network; in both cases, transfers flowed towards co-ethnics, kin and friends. This provides evidence that economic games, if designed to, can measure behavioural preferences that ostensibly operate in similar real-world contexts—in this case, motives for sharing with known members of the community.

The real-world resource transfer network data show that all else equal, resources preferentially flow from haves to have-nots: elderly, depressed, and food-insecure individuals are less likely to make resource transfers to others (presumably due to personal need), and those individuals with high grip strength are less likely to receive incoming transfers from others (presumably due to higher physical status and less need). A similar pattern also holds in the economic game data, but important differences emerge because constraints on behaviour are experimentally relaxed. In the allocation game, food-insecure individuals are just as likely as everyone else to make transfers to others; the same is true for depressed individuals in the allocation game. By ensuring that these individuals have resources (coins) that they can transfer to—or leave for—others if they wish, RICH games permit deciders to act in accordance with their private preferences.

Even more apparent are the effects of *recipient* characteristics as resource constraints are experimentally lifted. In the allocation game, deciders prefer to transfer coins to those individuals who are unable to work. As resource constraints are relaxed further in the taking game, we see preferences to leave coins for those individuals who cannot work, individuals with food insecurity and indigenous individuals (who generally are, and are perceived to be, living under tougher circumstances). Moreover, in the taking game, deciders prefer to take coins from those high in material wealth and those with high grip strength. Similar patterns abound in the costly reduction game, where elderly, food-insecure and indigenous individuals are less likely to be punished, and individuals with high material wealth are more likely to be punished (see also [65]). In the 'real world', where agency is constrained, cases where the poor or weak punish the wealthy or strong might be rare, and thus harder to detect with observational methods. In summary, these findings show that the relaxation of constraints on behaviour can allow individuals increasing

agency with which to act on their private preferences. The extent to which individuals in this dataset *prefer* to help marginalized members of their community would be overlooked by restricting the study design to real-world transfers alone.

In general, RICH games have shown high ecological validity: that is, they map onto the same dyadic variables that structure empirical resource transfers. However, they also provide deciders with more freedom to act on their private-world preferences, at least when the game parameters are set such that they relax real-world constraints, like resource availability. C.T.R. clarified this pattern in post-game interviews: the majority of respondents, even those in relative marginalization, saw the allocation game, and especially the taking game, as an opportunity to 'do the right thing' and give to those most in need (see also [16]). It is only through the combination of experimental and self-report data that we can reveal the disconnect between what participants wish to do and what they can actually do given their circumstances and competing obligations to friends, kin and members of their own household (see [72,73] for related discussion).

# 4. How do we better design games to test our research questions?

The four studies discussed above are not flawless examples of how to best use economic games, but they are instructive. We have learned from their shortcomings and surprises, and these lessons may be useful for others who wish to use economic games to measure private-world or real-world preferences. As we identified in §1, some primary considerations for researchers running economic games should be: (i) the match between research questions and experimental design, (ii) the extent to which researchers wish to make inferences about real-world versus private-world preferences, and (iii) whether the experimental task will be comprehensible to participants—which is especially relevant when exporting an economic game from one cultural context to another. We highlight different strategies for addressing these considerations below. We also discuss these strategies in light of replicability, emphasizing the importance of conceptual replication.

## 4.1. Consideration 1: research questions first, game design second

To avoid a common criticism of the use of economic games, we recommend formulating hypotheses or research questions before selecting methodological tools. It may be tempting to use classical economic games because they have been used by many other researchers—e.g. because they are 'standardized' [9] and 'paradigmatic' [9,14]. However, without consideration of how theory motivates the choice of method, it may be difficult to interpret experimental results, and this may encourage *post hoc* theorizing. Best practice includes being deliberate in one's choice of methods: methods should be appropriate for the research questions at hand, and this may require modifying existing tools, including classical economic games. For example, if you are interested in eliciting private-world levels of trust between ethnic groups, economic games may provide different insights than do self-report methods (see [74,75] for relevant discussion). However, if you hypothesize that threat affects trust between ethnic groups, the classical Trust Game [76] may not sufficiently capture this aspect of the real world; instead, consider modifying the game design so that threat is a salient part of the game [77].

## 4.2. Consideration 2: what does 'real world' mean, and do you want to measure real-world preferences?

As we illustrated in §§2 and 3, games can reveal private-world or real-world preferences. If you wish to draw inferences about real-world preferences, then relevant features of the real world need to be represented in the game—that is, the game must be *ecologically valid*, at least with respect to the context to which you wish to extrapolate (the *focal context*) [78]. To ensure the focal context is salient to participants, we recommend including it in the game design, a technique that has been successfully used by a number of researchers in a variety of locations [16,26,28,79,80]. For example, if the focal context is a cooperative institution, consider explicitly framing the experiment in terms of that institution [24] and/or mimicking the institution in the design of the game [79]. If real-world relationships or rare or hard-to-observe interactions are part of your research focus, consider revealing some characteristics of the recipients to the decider—for example, identifying recipients by name, photo [16,32,33], or group or community membership [52]. If everyday generosity is of interest, rather than giving participants a windfall of free money for game play, consider designing the game such that deciders are required to earn the money they give away (see [81] for useful discussion).

By establishing a principled means of quantitatively or qualitatively measuring the focal context, it becomes easier to compare experimental results with participants' real-world behaviour. Observational [21] and survey data are the most readily available options, but other data sources (e.g. audits of local records [59]) may be relevant as well. Though the magnitudes of effects may differ when comparing analyses of game data with analyses of observational or survey data, the direction of effects should be consistent if the game design reflects the focal context [17].

We also recommend including post-game questions asking participants what they thought the game was about [13,16,25,79]. If participants respond that the game was about the focal context, these data support the effectiveness of your manipulation (that is, its construct validity); if the focal context was invoked in the game design rather than in the game instructions, participants should likewise mention that context in their responses. Sometimes, unforeseen interpretations of the game that emerge in post-game questions will be frequent enough to warrant *post hoc* consideration (e.g. when games remind participants of putting money in the charity box at a temple, deciders might give more [54]).

If instead of real-world preferences you are attempting to measure *private*-world preferences, be deliberate about which constraints on real-world behaviour you will experimentally relax. For example, providing coins for game play can relax resource constraints—revealing, for example, reciprocity or need-based generosity— while making decisions confidential can relax reputational concerns—revealing, for example, punitiveness. Measures of the focal context and post-game questions can work in synergy with these manipulations. Post-game interviews can function like a manipulation check; participants may even tell you if they are acting on their private-world preferences (see C.T.R.'s experience in §3.3 for an example).

## 4.3. Consideration 3: know the community

Time spent on the ground—that is, observing and interacting with community members—permits researchers to substantially improve their game design and to more easily interpret their results. Time spent with a community can give you a sense for which methods will be most appropriate to test your research question. For example, to make the game relevant for participants, you may find it helpful to use a currency other than money (e.g. [82,83]). Furthermore, the more you know about the community, the better you will be able to interpret your results—something that is true regardless of the population with whom you are collaborating, even if you are studying undergraduates (see [3] for an example).[2] If you are limited in the time you can spend with a community, consider reading additional materials about the area to learn about the local context.[3]

Instead of relying solely on post-game questions, researchers can also assess how their game design reflects (or does not reflect) daily life by conducting a pilot with a small sample of individuals—a sample as representative as possible of the backgrounds, genders, ages, etc. of the community with whom they are collaborating. We recommend paying special attention to what pilot participants find perplexing or laughable (see [85] for an instructive example), asking them what they thought the game was about (e.g. [13,25]) and exploring what they found confusing. A.C.P. piloted her games; in her experience, pausing to consider alternative framings and protocols after each pilot participant substantially improved the comprehensibility of her game. If possible, we recommend piloting with members of a *different* community, but one similar to the focal community in the parameters of interest (e.g. living in the same region, growing the same crops, composed of people with similar ethnic backgrounds and similar rates of migration, etc.). Piloting in the focal community may not be desirable if (i) you intend to interview every household in the community (as did C.T.R. and M.M.G.), or (ii) you fear that participants may talk to one another about your game before they play (A.C.P.'s concern; see [72,73] for similar concerns). Per (ii), consider a pilot community that interacts infrequently with the focal community (A.C.P. picked communities living on different roads in the same region).

## 4.4. What about replicability?

In the past decade, conversations about replicability have come to the forefront in the social sciences. Replicability refers to whether an experimental result can be reproduced by other researchers who use the same experimental protocol with a sample of individuals drawn from the same population [86]; different criteria can be used to establish replication, but under the least stringent criterion, a successful

[2]Of course, such an approach requires some patience and an eye to ethics (see [84] for a helpful guide).

[3]We recommend reading the previous writings of ethnographers, and if possible, local community members themselves, describing the local cultural context; if these sources are not available, pieces by journalists or non-governmental organizations can also be helpful.

replication is one that finds effects in the same direction as those of the original experiment (see [87] for examples).

Classical economic games have proven reliable and generally yield replicable results within Western populations (e.g. [88]). They have also played a central role in some recent demonstrations that common Dictator and Ultimatum Game findings from Western student samples do not replicate across populations [31,60,89]. However, this failure to replicate has two interpretations. One is that the underlying phenomena, for example private-world or real-world preferences, vary fundamentally across populations; we do not doubt that there is genuine variation in the preferences underlying economic game play. That said, the second interpretation is methodological: perhaps, the validity of economic games—especially games featuring minimalistic instructions, anonymous recipients and money—is compromised in some populations, such as those that are less market integrated and predominantly interact with known individuals, not with strangers [11,13]. While designing games to enhance their local relevance, as we have recommended here, may boost their ecological validity [16,74], these alterations undercut strict replicability. Nonetheless, economic games tailored to particular contexts lend themselves to 'conceptual replications': if researchers have an *a priori* reason to think that participants' decisions will vary in a specific way, or will not vary, across different contexts, these theoretical predictions can be rigorously tested by triangulation, using slightly different methodological approaches across sites [90]. In other words, the economic games we have described produce results that are generalizable in a qualitative, but not necessarily quantitative, sense. At the same time, economic games can maximize other important domains of scientific research, such as ecological validity (as discussed in §1; [6,8]), and construct validity (as discussed in §4.2; [78,84,90]).

# 5. Conclusion

Focused on standardization [9] and parallels to bargaining in market societies [2], classical economic games will not map onto all real-world contexts or be comprehensible to all peoples. Nevertheless, by selectively altering the design of economic games—such as their framing, the presence or absence of anonymity, or the number of potential recipients—researchers can tailor this broad experimental method to their questions and to the communities with whom they collaborate. Notably, economic games can reveal individuals' private, minimally constrained preferences, or their preferences given real-world constraints; as our case study from Colombia illustrates, subtle changes in game design can elicit one or the other. Researchers can be more confident they are tapping real-world, as opposed to private-world, preferences if they explicitly include the focal context in the game design or experimental framing, supplement experimental data with observational or survey data and include post-experiment follow-up questions asking participants what they thought the game was about.

In §4, we provided recommendations for how and when to use economic games. There are situations in which observational or survey-based methods may be more appropriate for answering research questions, but economic games are especially useful for selectively relaxing real-world constraints to reveal private preferences, and for studying rare social interactions that may not be salient during an interview. Games tailored to research questions and to local research contexts can facilitate conceptual replication, a component of larger replication efforts. Furthermore, pilot runs of games and the inclusion of post-game questions can provide substantial qualitative data about local social interactions and relationships, enhancing the insights that economic games provide.

As with all social science methods, economic games are not without their limitations, but many of these limitations can be addressed with careful experimental design. While we do not recommend the use of economic games in isolation, they can generate a great deal of value as part of a larger toolkit deployed to investigate questions about human social behaviour and its variability.

Ethics. C.T.R. obtained informed consent from each respondent and the community leader (when appropriate) prior to data collection. Because of limited literacy rates at the study site, informed consent was obtained verbally. All field protocols were approved by the Max Planck Institute for Evolutionary Anthropology, Department of Human Behavior, Ecology and Culture, and declared exempt from additional IRB oversight. For details about the other studies discussed, see [16,33,52].

Data accessibility. Data and code are available at: www.github.com/ctross/preferencesandconstraints. For data and code from the other studies discussed, see [16,33,52].

Authors' contributions. A.C.P., M.M.G., B.G.P. and C.T.R. wrote the manuscript. C.T.R. collected and analysed the Colombian data. All authors gave final approval for publication.

Competing interests. We have no competing interests.

Funding. C.T.R. received support from the Max Planck Institute for Evolutionary Anthropology. For funding details for the other studies discussed, see [16,33,52].

**Acknowledgements.** Thanks to five anonymous reviewers for feedback on earlier versions of this manuscript, to Jeremy Koster for statistical advice and to Richard McElreath for helpful discussion.

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
