## [Reviewer comments · Royal Society Open Science]

Review History

RSOS-192090.R0 (Original submission)

Review form: Reviewer 1

Is the manuscript scientifically sound in its present form?

Yes

Are the interpretations and conclusions justified by the results?

Yes

Is the language acceptable?

Yes

Do you have any ethical concerns with this paper?

No

Have you any concerns about statistical analyses in this paper?

No

Recommendation?

Accept with minor revision (please list in comments)

Comments to the Author(s)

This is an insightful paper highlighting the value of economic games in human behaviour research, while acknowledging the limitations of these methods. A particular strength is the focus on using non-standard economic games in anthropological societies. I enjoyed reading this paper, and in my opinion, it will make a meaningful contribution to the growing methodological literature economic games. I am happy to recommend publication, with a few minor revisions.

- P.3, line 42: The sentence “Furthermore, researchers can design...” is unclear. Although I understand what you are getting at, this sentence would be strengthened by clarifying what is meant by reputational damage and resource limitations
- Throughout the text the authors reference “private preferences”. It would be of value to state why researching private preferences is important – how does it enhance our understanding of human behavior? I believe a short discussion on this would also strengthen the argument for the value of using economic games.
- P.4, line 55: Here it states that “in most economic games, researchers provide participants with money, often the equivalent of a day’s wages or more”. At this point in the paper, the focus on non-WEIRD populations has not been introduced, and therefore this statement should be applicable to all use of economic games (most of which is in WEIRD populations, where they are paid much less than this – see Larney, Rotella, & Barclay, 2019 for examples). Thus, this statement should be clarified to accurately reflect the use of all economic games, or add this caveat.
- Very small point – but you refer to a ‘two-colored die’ in a black and white figure, while you have other colored figures in the manuscript where color is less necessary.

Review form: Reviewer 2

Is the manuscript scientifically sound in its present form?

Yes

Are the interpretations and conclusions justified by the results?

Yes

Is the language acceptable?

Yes

Do you have any ethical concerns with this paper?

No

Have you any concerns about statistical analyses in this paper?

No

Recommendation?

Accept as is

Comments to the Author(s)

This is a very well-written and carefully reasoned paper. I’ve struggled thinking through these topics over the years, and I learned a lot from it. Thanks to the authors!

I think this should be accepted in basically its present form. I have only two minor suggestions for the main text figures that the authors might wish to consider. There also appears to be a minor error with the Supplement – is there really an ‘education’ variable?

1.

Could Fig 1 be improved such that the recipient's (non)decision is visualized? Not sure the figure is needed if the contrast isn’t more clear. Something should be coming back in the Ultimatum game, right? Or a thumbs up/thumbs down?

2.

Fig 5 is hard to digest. Perhaps add some guidance on how to interpret C-E especially (since A & B are interpreted in the text). Maybe reduce the thickness of the ties?

3.

From Supplement section interpreting model results:

“In the first column, top block, we see that food insecure, educated, and depressed individuals are significantly less likely to report transferring food or money to other community members”

Educated? I see no such variable on the Figure, and I don't see it introduced in the text above.

Review form: Reviewer 3

Is the manuscript scientifically sound in its present form?

Yes

Are the interpretations and conclusions justified by the results?

Yes

Is the language acceptable?

Yes

Do you have any ethical concerns with this paper?

No

Have you any concerns about statistical analyses in this paper?

No

Recommendation?

Accept with minor revision (please list in comments)

Comments to the Author(s)

I thought this was a very interesting paper, tackling the knee-jerk response that many anthropologists and psychologists have to economic games (but they are not ecologically valid!) by thinking hard about the ways that they can be useful, and their obvious limitations turned into virtues. To put this another way around, it is a paper about what economic games can be used to tell you and what they cannot. I liked the emphasis on not just using the standard game paradigms because they are standard, but instead trying to think creatively given one's research objectives.

Major points

There were a few things I found unclear.

First, there seemed to be a theme here, which was that economic games are good to use where there is a (truly experimental) independent variable, since it might reveal the causal significance of that IV (e.g. kinship or ethnicity of recipient) in the lives and thought of the people under study; but it is more problematic to use the brute level of cooperation in a game with no IV as if it represented something meaningful (e.g. how prosocial are the Gnau? Well, they contribute 46% on the DG). This is meaningless since you don't know how they are thinking about the game, how it was framed for them, how it relates to anything they do in life, etc. This seems like a useful distinction (estimating the effect of an IV vs. naively interpreting a level), but the authors never

come quite out and say anywhere that this is their meaning, or that they are against the latter. I see the naive interpretation of brute levels in the literature everywhere. So the authors could be clearer and more explicit on this point (if indeed this is their meaning).

Constraints. The authors make great play of the fact the standardized game designs can be used to eliminate some of the constraints of studying real-world behaviour in the classic observational way. This makes sense to me. However, the examples given of 'constraints' are heterogeneous and confusing. One obvious constraint is resources: in the game, by providing the resources, you are removing the constraint in the real world that some participants might have none. But other constraints are listed as 'moral' or 'reputational concerns'. These seem to me to be constraints of another kind, that are NOT removed by doing a standardized games. Even if told that games are anonymous, I might behave AS IF there were reputational consequences, because that psychology is hard to turn off. And morality does not cease to constrain me necessarily within a game. So, in short, I think the notion that constraints are lifted in games and therefore pure private preferences are being evoked is problematic. Even my privatest private preferences still have moral constraints, since these are internalized, and certainly my game play is still morally constrained. My guess is that games lift some constraints and not others, and only an economist would think there is a 'real' private preference that exists and is prior to all kinds of social constraints. This needs clarifying.

Finally, the paper says that games can be used for two different paradigms. 1. To study private preferences when the constraints of the real world are lifted (the private-world approach); and 2. as a proxy for how people might behave in real situations (the real-world approach). The authors say that these two approaches are often conflated and unfairly criticized, but my concern would be: how do you tell which one you are doing? The actual methods seem much the same in the two cases; only the aims and interpretation differ. So how can we distinguish what it is we are learning about in doing any particular study, and what methodological rules should drive us if we want to know about the real-world or the private-world. (By the way, I think the idea that there is a private-world set of preferences completely unpenetrated by real-world constraints is itself rather naive, as mentioned above).

It would be helpful if the authors could clarify their statements on the three matters above. The paper feels like it may have been written in sections, and these issues somewhat arise from the sections having not been wholly integrated into an overall statement.

Decision letter (RSOS-192090.R0)

10-Mar-2020

Dear Dr Pisor

On behalf of the Editors, I am pleased to inform you that your Manuscript RSOS-192090 entitled "Preferences and constraints: The value of economic games for studying human behavior" has been accepted for publication in Royal Society Open Science subject to minor revision in accordance with the referee suggestions. Please find the referees' comments at the end of this email.

The reviewers and handling editors have recommended publication, but also suggest some minor revisions to your manuscript. Therefore, I invite you to respond to the comments and revise your manuscript.

- Ethics statement

- Data accessibility

<http://datadryad.org/submit?journalID=RSOS&manu=RSOS-192090>

- Competing interests

- Authors' contributions

- Acknowledgements

- Funding statement

Because the schedule for publication is very tight, it is a condition of publication that you submit

the revised version of your manuscript before 19-Mar-2020. Please note that the revision deadline will expire at 00.00am on this date. If you do not think you will be able to meet this date please let me know immediately.

If your manuscript is newly submitted and subsequently accepted for publication, you will be asked to pay the article processing charge, unless you request a waiver and this is approved by Royal Society Publishing. You can find out more about the charges at

<https://royalsocietypublishing.org/rsos/charges>. Should you have any queries, please contact openscience@royalsociety.org.

on behalf of Dr Simone Schnall (Associate Editor) and Essi Viding (Subject Editor)
openscience@royalsociety.org

Associate Editor Comments to Author (Dr Simone Schnall):

As you will see in the enclosed evaluations, the reviewers were very positive about your paper. I agree that it could make an important contribution to the growing literature using economic games. I therefore ask you to address all their comments, which mostly are minor, given how positive the reviews were. Reviewer 3 made some more substantive points, which will be important to consider. I look forward to your revision.

Reviewer comments to Author:

Reviewer: 1

Comments to the Author(s)

This is an insightful paper highlighting the value of economic games in human behaviour research, while acknowledging the limitations of these methods. A particular strength is the focus on using non-standard economic games in anthropological societies. I enjoyed reading this paper, and in my opinion, it will make a meaningful contribution to the growing methodological literature economic games. I am happy to recommend publication, with a few minor revisions.

- P.3, line 42: The sentence “Furthermore, researchers can design...” is unclear. Although I understand what you are getting at, this sentence would be strengthened by clarifying what is meant by reputational damage and resource limitations
- Throughout the text the authors reference “private preferences”. It would be of value to state why researching private preferences is important – how does it enhance our understanding of human behavior? I believe a short discussion on this would also strengthen the argument for the value of using economic games.
- P.4, line 55: Here it states that “in most economic games, researchers provide participants with money, often the equivalent of a day’s wages or more”. At this point in the paper, the focus on non-WEIRD populations has not been introduced, and therefore this statement should be applicable to all use of economic games (most of which is in WEIRD populations, where they are paid much less than this – see Larney, Rotella, & Barclay, 2019 for examples). Thus, this statement should be clarified to accurately reflect the use of all economic games, or add this caveat.
- Very small point – but you refer to a ‘two-colored die’ in a black and white figure, while you have other colored figures in the manuscript where color is less necessary.

Reviewer: 2

Comments to the Author(s)

This is a very well-written and carefully reasoned paper. I’ve struggled thinking through these topics over the years, and I learned a lot from it. Thanks to the authors!

I think this should be accepted in basically its present form. I have only two minor suggestions for the main text figures that the authors might wish to consider. There also appears to be a minor error with the Supplement – is there really an ‘education’ variable?

1.

Could Fig 1 be improved such that the recipient's (non)decision is visualized? Not sure the figure is needed if the contrast isn't more clear. Something should be coming back in the Ultimatum game, right? Or a thumbs up/thumbs down?

2.

Fig 5 is hard to digest. Perhaps add some guidance on how to interpret C-E especially (since A & B are interpreted in the text). Maybe reduce the thickness of the ties?

3.

From Supplement section interpreting model results:

“In the first column, top block, we see that food insecure, educated, and depressed individuals are significantly less likely to report transferring food or money to other community members”

Educated? I see no such variable on the Figure, and I don't see it introduced in the text above.

Reviewer: 3

Comments to the Author(s)

I thought this was a very interesting paper, tackling the knee-jerk response that many anthropologists and psychologists have to economic games (but they are not ecologically valid!) by thinking hard about the ways that they can be useful, and their obvious limitations turned into virtues. To put this another way around, it is a paper about what economic games can be used to tell you and what they cannot. I liked the emphasis on not just using the standard game paradigms because they are standard, but instead trying to think creatively given one's research objectives.

Major points

There were a few things I found unclear.

First, there seemed to be a theme here, which was that economic games are good to use where there is a (truly experimental) independent variable, since it might reveal the causal significance of that IV (e.g. kinship or ethnicity of recipient) in the lives and thought of the people under study; but it is more problematic to use the brute level of cooperation in a game with no IV as if it represented something meaningful (e.g. how prosocial are the Gnau? Well, they contribute 46% on the DG). This is meaningless since you don't know how they are thinking about the game, how it was framed for them, how it relates to anything they do in life, etc. This seems like a useful distinction (estimating the effect of an IV vs. naively interpreting a level), but the authors never come quite out and say anywhere that this is their meaning, or that they are against the latter. I see the naive interpretation of brute levels in the literature everywhere. So the authors could be clearer and more explicit on this point (if indeed this is their meaning).

Constraints. The authors make great play of the fact the standardized game designs can be used to eliminate some of the constraints of studying real-world behaviour in the classic observational way. This makes sense to me. However, the examples given of 'constraints' are heterogeneous and confusing. One obvious constraint is resources: in the game, by providing the resources, you are removing the constraint in the real world that some participants might have none. But other constraints are listed as 'moral' or 'reputational concerns'. These seem to me to be constraints of another kind, that are NOT removed by doing a standardized games. Even if told that games are anonymous, I might behave AS IF there were reputational consequences, because that

psychology is hard to turn off. And morality does not cease to constrain me necessarily within a game. So, in short, I think the notion that constraints are lifted in games and therefore pure private preferences are being evoked is problematic. Even my privatest private preferences still have moral constraints, since these are internalized, and certainly my game play is still morally constrained. My guess is that games lift some constraints and not others, and only an economist would think there is a 'real' private preference that exists and is prior to all kinds of social constraints. This needs clarifying.

Finally, the paper says that games can be used for two different paradigms. 1. To study private preferences when the constraints of the real world are lifted (the private-world approach); and 2. as a proxy for how people might behave in real situations (the real-world approach). The authors say that these two approaches are often conflated and unfairly criticized, but my concern would be: how do you tell which one you are doing? The actual methods seem much the same in the two cases; only the aims and interpretation differ. So how can we distinguish what it is we are learning about in doing any particular study, and what methodological rules should drive us if we want to know about the real-world or the private-world. (By the way, I think the idea that there is a private-world set of preferences completely unpenetrated by real-world constraints is itself rather naive, as mentioned above).

It would be helpful if the authors could clarify their statements on the three matters above. The paper feels like it may have been written in sections, and these issues somewhat arise from the sections having not been wholly integrated into an overall statement.

Author's Response to Decision Letter for (RSOS-192090.R0)

See Appendix A.

Decision letter (RSOS-192090.R1)

Dear Dr Pisor,

It is a pleasure to accept your manuscript entitled "Preferences and constraints: The value of economic games for studying human behavior" in its current form for publication in Royal Society Open Science.

on behalf of Dr Simone Schnall (Associate Editor) and Essi Viding (Subject Editor)
openscience@royalsociety.org

Associate Editor Comments to Author (Dr Simone Schnall):

Associate Editor

Comments to the Author:

Thank you for preparing such a thorough revision that carefully addressed all the reviewers' comments. The novel contribution of your work now comes out even more clearly, and I'm sure it will be a valuable resource to many of our readers. It is therefore my pleasure to accept your manuscript for publication.

Appendix A

Dear Dr Schnall,

We are writing to resubmit the manuscript “Preferences and constraints: The value of economic games for studying human behavior” for your consideration. Thanks to you and the three reviewers for your comments; your input has helped us improve the clarity of our manuscript.

The most substantial changes to the revised manuscript involve clarification of what we mean by “private preferences” and why they are important to study, per comments from two reviewers. Other, smaller changes have added detail and explanation and removed errors. All line numbers below refer to the “clean” version of the manuscript, with changes accepted. Outside of the review process, a conversation with a colleague inspired Cody to add dyadic random effects to the statistical models as a robustness check. The SOM includes details on this new approach in the Methods section; the SOM and main text have also been lightly modified to reflect the new modeling results (including an updated Figure 6). Results were qualitatively unchanged by the robustness check.

We look forward to your assessment of our revised piece.

Cordially,

Anne Pisor, Cody Ross, Matthew Gervais, and Benjamin Purzycki

Associate Editor Comments to Author (Dr Simone Schnall):

As you will see in the enclosed evaluations, the reviewers were very positive about your paper. I agree that it could make an important contribution to the growing literature using economic games. I therefore ask you to address all their comments, which mostly are minor, given how positive the reviews were. Reviewer 3 made some more substantive points, which will be important to consider. I look forward to your revision.

We thank you for your assessment, and the reviewers for their time and constructive feedback. We respond point-wise to each of the substantive points below.

Reviewer comments to Author:

Reviewer: 1

Comments to the Author(s)

This is an insightful paper highlighting the value of economic games in human behaviour research, while acknowledging the limitations of these methods. A particular strength is the focus on using non-standard economic games in anthropological societies. I enjoyed reading this paper, and in my opinion, it will make a meaningful

contribution to the growing methodological literature economic games. I am happy to recommend publication, with a few minor revisions.

Thanks for the positive feedback.

- P.3, line 42: The sentence “Furthermore, researchers can design...” is unclear. Although I understand what you are getting at, this sentence would be strengthened by clarifying what is meant by reputational damage and resource limitations.

We have now clarified our previous statement by rephrasing it (on lines 42-46), as follows:

“Furthermore, researchers can design games such that they minimize (though not eliminate) real-world constraints—for example, those posed by resource limitations (individuals simply may not have extra resources to give to others), or reputational consequences of real-world behavior (individuals may have preferences to be stingy that they are unable to express in daily life)—revealing participants’ private preferences in a way that observational and interview data may not.”

- Throughout the text the authors reference “private preferences”. It would be of value to state why researching private preferences is important – how does it enhance our understanding of human behavior? I believe a short discussion on this would also strengthen the argument for the value of using economic games.

This is a good point. We have now added some brief comments concerning the relevance of preferences to our understanding of human decision-making. In the context of interventions, for example, it is often helpful to understand preferences apart from contextual limitations. If people refrain from a given behavior (e.g., the over-use of chemical pesticides) because of internal preferences against that behavior, rather than from lack of access to the resources needed to engage in that behavior (e.g., supply quotas), then results of an intervention (deregulation) might be very different than for another group with a different set of preferences/constraints. Measurement of private preferences can thus play an essential role in understanding how individuals would behave *if they could*, and thus in predicting the results of interventions or actions (see our revisions on lines 116-118 and 123-126).

- P.4, line 55: Here it states that “in most economic games, researchers provide participants with money, often the equivalent of a day’s wages or more”. At this point in the paper, the focus on non-WEIRD populations has not been introduced, and therefore this statement should be applicable to all use of economic games (most of

which is in WEIRD populations, where they are paid much less than this – see Larney, Rotella, & Barclay, 2019 for examples). Thus, this statement should be clarified to accurately reflect the use of all economic games, or add this caveat.

Another good point: we hadn't noticed the discrepancy before. Using the citation recommended here (which we had not yet seen), we have modified lines 57-59 to state:

"In most economic games, researchers provide participants with money – ranging from the equivalent of a few minutes' work to a day's wages or more [14] – and ask them to decide how much to keep for themselves and how much to allocate to third parties (the recipients)."

- Very small point – but you refer to a 'two-colored die' in a black and white figure, while you have other colored figures in the manuscript where color is less necessary.

We have clarified in the relevant figure caption (Fig. 2).

Reviewer: 2

Comments to the Author(s)

This is a very well-written and carefully reasoned paper. I've struggled thinking through these topics over the years, and I learned a lot from it. Thanks to the authors!

We're glad to hear it! Thanks for the feedback here.

I think this should be accepted in basically its present form. I have only two minor suggestions for the main text figures that the authors might wish to consider. There also appears to be a minor error with the Supplement – is there really an 'education' variable?

This text was erroneously left in from an older version of the model. We have revised the Supplementary Materials and checked that we didn't have any similar mistakes.

1. Could Fig 1 be improved such that the recipient's (non)decision is visualized? Not sure the figure is needed if the contrast isn't more clear. Something should be coming back in the Ultimatum game, right? Or a thumbs up/thumbs down?

Good suggestion! We have added visuals to Figure 1 to clarify the difference between the Dictator and Ultimatum games.

2. Fig 5 is hard to digest. Perhaps add some guidance on how to interpret C-E especially (since A & B are interpreted in the text). Maybe reduce the thickness of the ties?

We can't reduce the line thickness much, given programming constraints. In the case of the leaving game, though, the density is so high that it will look like this for any line thickness. The high density, however, is really the key point of the sub-figure. To give more guidance, we have revised the caption as below:

“Network structure of (A) social relationships, (B) resource transfers, and (C-E) RICH game data. Afrocolombians are plotted in black and Emberá in blue. Points (also called vertices) represent unique individuals and are scaled by material wealth. Ties (also called edges) represent social connections or resource/experimental transfers between a pair of individuals; both unidirectional and bidirectional ties are depicted. Frame (A) illustrates the social relationships between individuals and illustrates the role of ethnicity and wealth similarity in the structure of these relationships. Frame (B) shows the flow of food and money transfers and illustrates a rather low density of connections. In frame (C), a higher density of transfers in the RICH game becomes apparent, as every decider has money available to give should they choose to. Giving remains more common between dyads of the same ethnicity. In frame (D), deciders have enough coins to allocate to all recipients, and the resulting density of transfer ties increases. The abundance of ties reflects that most deciders left money for (i.e., failed to take money from) most recipients. Finally, in frame (E), we see that comparatively few deciders paid to reduce recipients.”

3. From Supplement section interpreting model results:

“In the first column, top block, we see that food insecure, educated, and depressed individuals are significantly less likely to report transferring food or money to other community members”

Educated? I see no such variable on the Figure, and I don't see it introduced in the text above.

This was a typo. It should have said “elderly”. Fixed. We have also re-read the whole Supplementary Materials and fixed up any similar issues.

Reviewer: 3

Comments to the Author(s)

I thought this was a very interesting paper, tackling the knee-jerk response that many anthropologists and psychologists have to economic games (but they are not

ecologically valid!) by thinking hard about the ways that they can be useful, and their obvious limitations turned into virtues. To put this another way around, it is a paper about what economic games can be used to tell you and what they cannot. I liked the emphasis on not just using the standard game paradigms because they are standard, but instead trying to think creatively given one's research objectives.

Thanks for your assessment – it sounds like we definitely got our message across.

Major points

There were a few things I found unclear.

First, there seemed to be a theme here, which was that economic games are good to use where there is a (truly experimental) independent variable, since it might reveal the causal significance of that IV (e.g. kinship or ethnicity of recipient) in the lives and thought of the people under study; but it is more problematic to use the brute level of cooperation in a game with no IV as if it represented something meaningful (e.g. how prosocial are the Gnau? Well, they contribute 46% on the DG). This is meaningless since you don't know how they are thinking about the game, how it was framed for them, how it relates to anything they do in life, etc. This seems like a useful distinction (estimating the effect of an IV vs. naively interpreting a level), but the authors never come quite out and say anywhere that this is their meaning, or that they are against the latter. I see the naive interpretation of brute levels in the literature everywhere. So the authors could be clearer and more explicit on this point (if indeed this is their meaning).

As a team, we extensively discussed being more explicit on this point during the writing process. For diplomatic reasons, we decided to go no further than simply providing evidence against the utility of (what is referred to here as) the brute level approach. We aimed to guide the reader toward such a conclusion. However, we do lean into our claim a little more now on lines 94-94, as per this comment. We similarly lean in on lines 330-334.

Constraints. The authors make great play of the fact the standardized game designs can be used to eliminate some of the constraints of studying real-world behaviour in the classic observational way. This makes sense to me. However, the examples given of 'constraints' are heterogeneous and confusing. One obvious constraint is resources: in the game, by providing the resources, you are removing the constraint in the real world that some participants might have none. But other constraints are listed as 'moral' or 'reputational concerns'. These seem to me to be constraints of another kind, that are NOT removed by doing a standardized games. Even if told that games are anonymous, I might behave AS IF there were reputational consequences, because that psychology is

hard to turn off. And morality does not cease to constrain me necessarily within a game. So, in short, I think the notion that constraints are lifted in games and therefore pure private preferences are being evoked is problematic. Even my privatest private preferences still have moral constraints, since these are internalized, and certainly my game play is still morally constrained. My guess is that games lift some constraints and not others, and only an economist would think there is a 'real' private preference that exists and is prior to all kinds of social constraints. This needs clarifying.

This is a good point. There was also some heterogeneity in word-usage across sections in our submission. In some cases, we treated private-world preferences as subject to “minimal constraints” and other sections treated them as being “without constraints.” We have now harmonized the message across the paper to “minimal constraints,” as that was our intended meaning. We agree with your point that people internalize moral norms and that there is no such thing as “private preferences” in the absence of socially transmitted cultural information that folks internalize. We have now laid this out as an explicit contribution to private preferences when we introduce the concept in Section 2.1 (lines 118-123). We also emphasize in the same sentence that private preferences are not independent of culture.

Finally, the paper says that games can be used for two different paradigms. 1. To study private preferences when the constraints of the real world are lifted (the private-world approach); and 2. as a proxy for how people might behave in real situations (the real-world approach). The authors say that these two approaches are often conflated and unfairly criticized, but my concern would be: how do you tell which one you are doing? The actual methods seem much the same in the two cases; only the aims and interpretation differ. So how can we distinguish what it is we are learning about in doing any particular study, and what methodological rules should drive us if we want to know about the real-world or the private-world. (By the way, I think the idea that there is a private-world set of preferences completely unpenetrated by real-world constraints is itself rather naive, as mentioned above).

We addressed this point in Section 4.2, which makes explicit recommendations for how to tailor games to either real-world or private-world preferences. To drive this point home, we now rehash the main recommendations from that section in our Conclusion, lines 441-444.

Regarding the reviewer’s final point, we note that we do not hold that “there is a private-world set of preferences completely unpenetrated by real-world constraints.” Our general point on preference/constraints is quite moderate: it may sometimes be the case that a person might wish to behave in one way but can’t/won’t do so, due to some

kind of constraining social context—e.g., a wage laborer might harbor resentments towards a disparaging boss, but not publicly speak/act in a way that makes such resentments known, due to fear of reprisal. This leads to problems with anthropological inference based solely on observed behavior: a lack of observed conflict, for example, does not imply true harmony.

We do think that use economic games where the behavior of the focal is anonymous can relax these kinds of social/reputational constraints to some extent. We do not think that an anonymous game context “turns off” all of the psychological mechanisms concerning the social consequences of public behavior, but we do find that it can sometimes be consequential. We think that we are actually in substantial agreement with the reviewer, and we hope the edits described above make this more clear in the paper.

It would be helpful if the authors could clarify their statements on the three matters above. The paper feels like it may have been written in sections, and these issues somewhat arise from the sections having not been wholly integrated into an overall statement.

Thank you for flagging this. We have revisited the entire piece and made small changes throughout to harmonize our message.